# Plant Triterpenoids Regulate Endophyte Community to Promote Medicinal Plant *Schisandra sphenanthera* Growth and Metabolites Accumulation

**DOI:** 10.3390/jof7100788

**Published:** 2021-09-23

**Authors:** Chuan You, Dan Qin, Yumeng Wang, Wenyi Lan, Yehong Li, Baohong Yu, Yajun Peng, Jieru Xu, Jinyan Dong

**Affiliations:** 1Chongqing Key Laboratory of Plant Resource Conservation and Germplasm Innovation, School of Life Sciences, Southwest University, Chongqing 400715, China; youchuan@email.swu.edu.cn (C.Y.); qindan2017@email.swu.edu.cn (D.Q.); w928986123@email.swu.edu.cn (Y.W.); lanlalawy@email.swu.edu.cn (W.L.); 18008338842@email.swu.edu.cn (Y.L.); ba323575@email.swu.edu.cn (B.Y.); peng199595@email.swu.edu.cn (Y.P.); xjr960313@email.swu.edu.cn (J.X.); 2Key Scientific Research Base of Pest and Mold Control of Heritage Collection (Chongqing China Three Gorges Museum), State Administration of Cultural Heritage, Chongqing 400015, China

**Keywords:** endophyte, triterpenoid, *Schisandra sphenanthera*, SynCom, interactions, growth, metabolites

## Abstract

Beneficial interactions between endophytes and plants are critical for plant growth and metabolite accumulation. Nevertheless, the secondary metabolites controlling the feedback between the host plant and the endophytic microbial community remain elusive in medicinal plants. In this report, we demonstrate that plant-derived triterpenoids predominantly promote the growth of endophytic bacteria and fungi, which in turn promote host plant growth and secondary metabolite productions. From culturable bacterial and fungal microbial strains isolated from the medicinal plant *Schisandra sphenanthera*, through triterpenoid-mediated screens, we constructed six synthetic communities (SynComs). By using a binary interaction method in plates, we revealed that triterpenoid-promoted bacterial and fungal strains (TPB and TPF) played more positive roles in the microbial community. The functional screening of representative strains suggested that TPB and TPF provide more beneficial abilities to the host. Moreover, pot experiments in a sterilized system further demonstrated that TPB and TPF play important roles in host growth and metabolite accumulation. In summary, these experiments revealed a role of triterpenoids in endophytic microbiome assembly and indicated a strategy for constructing SynComs on the basis of the screening of secondary metabolites, in which bacteria and fungi join forces to promote plant health. These findings may open new avenues towards the breeding of high yielding and high metabolite-accumulating medicinal plants by exploiting their interaction with beneficial endophytes.

## 1. Introduction

Endophytes (fungi and bacteria) colonize the tissue or intercellular spaces of various plant organs, live in healthy plants, at least at a certain stage or even during the whole life cycle, and do not cause obvious plant diseases [1]. The endophytic microbiome plays an important role in different functions such as plant growth, development, signal network, gene expression, metabolic pathways and response to biological stress [2,3,4]. The microbial community compositions of root, stem and leaf were clearly distinguished from rhizosphere soil and nonrhizosphere [5]. Plants actively recruit and nurture a collection of microbes from the soil to live in the vicinity, on the surface and inside their tissues [1]. Endophytic microbes provide various essential nutrients (i.e., nitrogen and phosphorous) to the host species, which promote plant growth and increase environmental tolerance and phytopathogenic resistance. In return, the plant partners offer stable niches and photosynthetic productions to the endophytes [6]. However, few studies have reported on the secondary metabolites that influence the assemblies of the endophytic microbiome in plants.

Plants have evolved their special secondary metabolites for metabolism, growth, development, and resistance to drought, disease, and radiation in natural environments [7]. An increasing body of evidence highlights the strong influences of secondary metabolites on the assembly and function of the plant microbiome. A group of secondary metabolites is involved in shaping the plant microbiome in the plant kingdom, such as triterpene [8], coumarin [9], camalexin [10], glucosinolate [2], and benzoxazinoid [11]. Plant secondary metabolites can exert a wide spectrum of effects upon individual microbial strains by functioning as signaling molecules, nutrient sources, or toxins [12]. In return, microbes promote host plant growth and health through enhancement of nutrient uptake, inhibition of pathogens, and maintenance of community stability [9,13,14]. Triterpenoid derivatives have functions in plant defense, signaling, and antimicrobial activity. Recent studies demonstrated that triterpenoids can selectively regulate the growth of *Arabidopsis*
*thaliana* root bacteria from different taxa [8]. With respect to the microbiome, endophytes affect plant growth and development by enhancing nutrient uptake, producing indole acetic acid, and alleviating stresses [14]. Endophytic microbial strains form a stable community structure through cooperation and competition to survive in and colonize plants [15,16,17,18,19]. For example, previous studies on *A**. thaliana*, soybean, tomato, garlic, and *Salvia miltiorrhiza* demonstrated that microbial interactions play an essential role in plant growth, health, and metabolite accumulation through some mechanisms including increasing antibiotic secretion, promoting biofilm formation and growth, and influencing metabolic pathways [14,20,21,22,23,24,25].

There are about 58 species in the world, which are mainly distributed in Southeast Asia and North America, and 29 species in China of the Schisandraceae family. As a widely domesticated medicinal plant in China, the Schisandraceae family has a very important position in China’s traditional medicine [26]. Furthermore, numerous secondary metabolites with wide biological activity were isolated from Schisandraceae plants, among which lignans and triterpenoids are the predominant groups. Since 1970, more than 1000 terpene-types of schtriterpenoids/schnortriterpenoids have been reported to be produced by the plants of Schisandraceae family [27]. Our previous studies demonstrated that endophytic fungi can specifically utilize and transform host triterpenoids in vitro [28,29]. Thus, triterpenoids may play a role in shaping the endophytic microbiome in the plant. 

In this study, we collected a Schisandraceae plant species from a natural environment in Xichou county of Yunnan province, corresponding to *S. sphenanthera*, and isolated and identified the culturable bacterial and fungal strains across 20 samples from multiple organs (root, stem, leaf, and fruit) of the plant species. Through in vitro bioassays with purified compounds of triterpenoids, the selective growth modulation activities of triterpenoids toward endophytes in *S. sphenanthera* were identified. The functional screening of culturable strains proved that triterpenoid-promoted bacterial and fungal strains (TPB and TPF) are beneficial microbes of plants. Then, we investigated the interaction between representative bacteria and endophytic fungi in vitro. Germ-free *S. sphenanthera* were further established to elucidate the activities of the synthetic communities (SynComs) with the greatest effect on *S. sphenanthera* growth and metabolic accumulation.

## 2. Materials and Methods

### 2.1. Sampling of S. Sphenanthera Plants in the Natural Habitat 

Healthy *S. sphenanthera* were collected in June of 2019 at Xichou, Yunnan province, China (altitude: 1399 m, 23°38.114′, E104°79.155′). Samples were taken from five *S. sphenanthera* plants separated by a variable distance ranging from 1 to 10 m. The plants were chosen without regard to their age or size. Five individual plants were dug out, transported on ice to the laboratory, and stored at −80 °C. Sample fractionation into fruits, leaves, stems, and roots was performed within 12 hours after harvesting. Lastly, 500 segments from each plant organ were plotted for the isolation of endophytic bacteria and fungi.

### 2.2. Isolation, Purification, Identification, and Preservation of Culturable Bacteria and Fungi 

In order to isolate endophytic fungi and bacteria, the plants were surface sterilized for 3 min in 80% EtOH, followed by a second sterilization step for 3 min in 0.25% NaClO. After washing the plants five times with sterile water, the fruits, leaves, roots, and stems were cut into small pieces (about 2 × 2 mm) and placed onto PDA, M + PDA, and CMM media (Appendix A) [30]. Endophytic fungi were purified and isolated by the tip mycelium picking method. Similarly, endophytic bacteria were isolated from plants on TSB, TYG, YEM, TWYE, MYX, MM+MeOH, R2A, MMF, and GAM, and purified by dilution (Appendix A) [31]. The DNA of the purified strains was extracted and amplified by PCR using bacterium-specific primers (1494R, 27F) and fungus-specific primers (ITS1, ITS4). The products were purified and sequenced in two directions. The 16S rRNA or ITS rRNA sequences were obtained by splicing the upstream and downstream sequences. Blast comparison was carried out in the NCBI GenBank database to find representative sequences that had a greater than 97% similarity. In total, 127 bacterial isolates and 119 fungal isolates were successfully identified according to their morphology and their 16S rDNA and ITS rDNA sequences, respectively. Further phylogenic tree analysis of sequenced isolates was conducted with multiple sequence alignments constructed in Clustal W [32] using the neighbor-joining method in MEGA 7.0 and visualized with iTOL v4 [33]. All collected isolates from plants were subsequently frozen in 30% or 40% glycerol stock solutions at −80 °C for storage. 

### 2.3. Isolation of Main Triterpenoid Metabolites from S. sphenanthera

Samples of roots, stems, leaves, and fruits (10 kg) from five *S. sphenanthera* plants were cut into pieces and dried naturally. After drying, the plant organs were soaked in 20 L of acetone for 15 days. The extraction was repeated a total of five times to fully extract the metabolites in the plant. The acetone extract was evaporated under reduced pressure in a rotary evaporator to obtain the crude extract. After extraction, the crude extract was subjected to MCI (CHP20P, 75~150 µ; Mitsubishi Chemical Corporation) column chromatography (CC) eluted with a gradient of CH_3_OH–H_2_O (1:9–1:0, 25 mL/min, 12 h) to afford 10 fractions (Fr. A–J) based on the basis of TLC pattern. Fraction C was purified over silica gel (chloroform–EtOAc, 10:1) and Sephadex LH-20 (MeOH) to furnish compound S4 (2.1 g). Fraction E (6 g) was further purified by silica gel column chromatography and eluted with gradient mixtures of petroleum ether EtOAc and AcOH (6:4:0.1) to yield compound S1 (9.0 g) and S2 (3.6 g). Fraction K was subjected to Sephadex LH-20 column eluted with MeOH and recrystallized to obtain compound S3 (4.5 g). The purity of all the isolated triterpenoids was greater than 95%, and the purity of all isolated triterpenoids was determined by HPLC analysis, which was carried out using an Agilent L-2000 liquid chromatograph equipped with a Zorbax SB-C18 (Agilent Technologies, Santa Clara, California, 9.4 mm × 15 cm column), The detection wavelengths of compounds S1–S4 were 212 nm (LC-UV%), 217 nm (LC-UV%), 203 nm (LC-UV%) and 243 nm (LC-UV%).

### 2.4. In Vitro Assay for Selective Effect of Triterpenoids

In total, 27 bacterial isolates and 26 fungal isolates from diverse taxa were selected for bioassay. To quantify the effect of triterpenoids on growth of the endophytic bacteria in vitro, we designed a microplate assay with a 96-well flat-bottomed plate on the basis of Huang et al. and Stringlis et al. [8,9]. Each assayed compound (compounds S1–S4) was mixed at a ratio of 1:1 in weight, and then was dissolved with ethanol to prepare a mix solution with different concentrations. Solvent ethanol was used as a control. A single colony from each strain was picked and inoculated into a 1/2TSB broth (Tryptone, Soya Broth (Oxoid) 15 g/L) and cultured at 37 °C overnight until the density reached OD_600_ 0.5–1.0. The cultures were then diluted 20–40 fold with 1/10TSB medium depending on the growth rate of the individual bacterial strain, and the resulting broth was used for bioassay. The growth medium was supplemented with different concentrations of triterpenoid mixtures and EtOH only in 1/2TSB media, and growth (OD_600_) was monitored over 72 h in the dark at 37 °C. The selective effect of endophytic fungus growth by triterpenoids in vitro was tested in a 6-well flat-bottomed plate by subculturing a mycelium plug of 5 mm diameter on 1/5 liquid potato dextrose broth (PDB) containing different concentrations of triterpenoid mixtures for 12 days in the dark at 28 °C. The mycelium was centrifuged at 12,000 g to remove the medium, and growth was recorded by weighing the fresh mycelium weight at 12 days.

### 2.5. Functional Screening for Potentially Beneficial Microbes

The procedure was used to screen for potentially beneficial microbes from 170 bacterial isolates and 170 fungal isolates. Bacterial and fungal indole 3-acetic acid (IAA) production was measured by using Salkowski reagent, as described by Hussain et al. [34]. The 500 µL of bacterial and fungal inoculum was dispensed in 100 mL TSB or PDA broth alone, or supplemented with 1% tryptophan, respectively. Bacterial and fungal strains were grown in broth for 24 h and 7 days, respectively, at 28 °C. A 1 mL bacterial or fungal supernatant was mixed with 2 mL of Salkowski reagent (4.5 g of FeCl_3_ per liter in 10.8 M H_2_SO_4_) and incubated at room temperature for 30 min. The developed color was monitored at 535 nm by using a CECIL Aquarius CE 7200 spectrophotometer (UK).

To evaluate siderophore production, CAS medium (blue agar chrome azurol S (CAS), medium containing chrome azurol S (CAS), and hexadecyltrim ethylammonium bromide (HDTMA)) were used to determine the siderophore production of the selected strains, as described by Alexander et al. [35]. Then, 500 µL of bacterial and fungal inoculum was dispensed in 100 mL TSB or PDA broth containing no added Fe, respectively. Bacterial and fungal strains were grown for 24 h and 7 days, respectively, at 28 °C. Siderophore concentration in the bacterial or fungal supernatant was measured by mixing 100 µL of modified CAS assay solution with 100 µL supernatant in a 96-well plate. Absorbance was measured at 630 nm with a microplate reader. 

For the determination of the 1,1-diphenyl-2-picrylhydrazyl (DPPH) radical, 100 µL of an aqueous solution of the bacterial and fungal supernatant sample (control:100 µL of distilled water) was added to an ethanolic solution of DPPH (60 µM) in microwells, according to the method of Yokozawa et al. [36]. After incubation, the DPPH radical was determined using a microplate reader. 

For P–solubilizing activity assays, P content in the fermentation liquid of bacteria and fungi was established by Mo–Sb colorimetry, as described by Gyaneshwar et al. [37]. Then, 1 mL of an aqueous solution of the bacterial and fungal supernatant sample (control:1 mL of phosphate solution) was added to 1% ascorbic acid solution (1 mL), followed by 2 mL of molybdate solution added after 30 s and reacted for 10 min. Lastly, absorbance was measured at 720 nm with a microplate reader. 

### 2.6. Assessment of Dual Interactions in Culture Media

Dual interactions of the 27 bacterial species were tested in the solid TSB agar (Appendix A) [20]. A single colony from each strain was picked and then inoculated into a 1/2TSB broth and incubated overnight at 37 °C at 180 rpm. The optical density of bacterial cultures was adjusted to 0.1 at 600 nm. Eleven times 1 µL of these dilutions were inoculated in a diagonal row on both sides of a circular culture dish with TSB agar with a multichannel pipet, creating a V shape of increasingly close inoculation sites. The plates were sealed with Parafilm and incubated for 15 days at 20 °C. Colony diameters were measured on a line, orthogonal to the line dividing the V shape. The interaction index (II) was calculated as follows: II = [(D_25mm-EA_/D_10mm-EA_)/(D_25mm-CA_/D_10mm-CA_) − 1] ∗ 100% (with D_25mm_ and D_10mm_ representing the neighboring colony diameter at 25 and 10 mm horizontal distance to the V shape, respectively; and D_EA_ and D_CA_ being the bacterial strain on the V-shape of the experiment and control group, respectively). Dual interactions of 25 fungal species were tested in solid PDA agar (Appendix A) [38]. Two mycelial disks with a diameter of 5 mm were placed in the center of a PDA plate with a distance of 2 cm. After growth at 28 °C for 3 days, scans were performed using a Konica Minolta Bizhub C652 (Konica Minolta Business Solutions U.S.A., Inc., Ramsey, NJ, USA) and saved as JPEG images (3024 × 4032 px, file size ranged from 1.19 to 3.01 MB). Scans were analyzed using the National Institutes of Health Image J v1.4 measuring tool software (rsb.info.nih.gov/ij/index.html). II was calculated as follows: II = (S_EA_/S_CA_−1) ∗ 100% (where S_EA_ and S_CA_ represent the measured colony area of fungi on the plate of the experiment and control groups, respectively). Dual interactions of the 27 bacterial species and 25 fungal species were tested in the solid PDA agar (Appendix A) [39]. A mycelial disk with a diameter of 5 mm was placed at 2 cm from the edge of a PDA plate. The 27-species bacterial suspension was streaked across the middle of the plate. A control was set up for challenging fungi without bacteria. After growth at 28 °C for 12 days, the diameter of the fungus colonies was recorded. II was calculated as follows: II = (S_EA_/S_CA_−1) × 100% (where S_EA_ and S_CA_ represented the colony diameter of fungi in the experiment and control group, respectively). II values below or above zero indicated growth inhibition and facilitation, respectively, and they were expressed as percentage fold changes.

### 2.7. SynCom Construction and Functional Evaluation

Surface-sterilized seeds of *S. sphenanthera* with 95% alcohol and sodium hypochlorite were planted in the sterilized soil and cultured at 28 °C for 90 days (see Appendix A). The young stem segments were cut for the rapid clonal propagation of *S. sphenanthera*. The propagation medium was MS + BA (1.0 mg/L) + NAA (0.03 mg/L), and the rooting medium was 1/2MS + IBA (0.5 mg/L). After the aseptic seedlings of *S. sphenanthera* had taken root (see Appendix A), the plants with the same growth momentum were transplanted into the sterilized soil in a culture bottle (see Appendix A) [40]. Tissue cultures of *S. sphenanthera* were planted in a small bottle with sterilized soil, on the basis of Liu et al. [25]. Three plants were sown in each pot, and three pots were set up for each treatment. For the construction of the synthetic community (SynCom), a total of 52 isolates were selected from diverse taxa [14]. The strain composition of the six SynComs was based on Appendix A. Bacterial strains were cultivated in 400 mL of 1/2 TSB overnight at 37 °C. These bacteria were centrifuged at 3000× *g* for 10 min and resuspended in 10 mM MgCl_2_ solution. Prior to inoculation, OD_600_ of the bacterial solution was diluted to 0.02 (107 cells/mL). Fungal strains were cultivated in 400 mL of PDB at 28 °C for 10 days. The mycelium pieces were collected and dispersed in 10 mM MgCl_2_ solution containing stainless-steel beads (3 mm). Then, 5 g of mycelium was dissolved in 50 mL 1/2 MS medium for co-culture system inoculation. After that, 1 mL of bacteria and 50 mL of fungi were inoculated by syringe into the root-soil of three-week-old *S. sphenanthera* seedlings with the same growth conditions [41]. SynCom inoculants were prepared through mixing equal amounts of each re-suspended microbial strain. All plants were grown in growth chambers for 60 days under long-day conditions (day/night: 14/10 h, 26/22 °C; light intensity: 2000 LX). Control plants were only supplied with 1/2 MS [4]. After 40 days, the plants were taken out and washed to measure their growth and physiological indices, including plant height, biomass, and chlorophyll content, and the content of metabolites. The whole plants of *S. sphenanthera* were dried in an oven at 55 °C for 24 h, and the dry weight of the plants was measured as biomass. To measure chlorophyll content, 200 mg of fresh leaf was extracted overnight in 8 mL of 80% (*v*/*v*) acetone, and absorbance at 645 and 663 nm was measured [42].

### 2.8. SEM Studies

To confirm the infection of bacteria and fungi in plants, scanning electron microscopy (JSM5910, JEOL, Japan) was used. The main roots of *S. sphenanthera* in the CK, AB, AF, and ABF group were collected after 28 days of co-culture, disinfected with 95% alcohol for 1 min, and fixed in 2.5% (*v*/*v*) glutaraldehyde (in 0.1M phosphate buffer; pH7.4) for 12 h at 4 °C. The roots were washed three times with 0.1M phosphate buffer for 5 min and dehydrated in a graded ethanol series (30%, 50%, 70%, 80%, 90%, 95%, and 100%) for 30 min each. Then, the root of *S. sphenanthera* was broken with a sterile scalpel, and the sample was dried overnight in a sterile dryer. The total dried samples were then coated with gold ion and submitted for SEM analysis [43].

### 2.9. Measurement of Plant Metabolite Content

Total sugars, total triterpenoid, flavonoids, and polyphenols were determined to measure changes in plant metabolites. For determining total sugar content, 1 g of plant dried powder was ground with 50 mL water, then transferred into a centrifuge tube and centrifuged at 5000× *g* for 20 min at 4 °C. Then, the crude extract of total sugar was obtained through rotation–evaporation. The standard curve was analyzed with anhydrous glucose as the standard, and the content of total sugar in the crude extract was determined by classical phenol sulfuric acid spectrophotometry [44]

Total triterpenoid was measured on the basis of Sun et al. [45], using vanillin–acetic acid–perchloric acid spectrophotometry to detect total triterpenoid in plants. For determining total triterpenoid content, 1 g of plant dried powder was ground with 50 mL acetone, then transferred into a centrifuge tube and centrifuged at 5000× *g* for 20 min at 4 °C. Then, the crude extract of total triterpenoid was obtained through rotation–evaporation. Of pure petroleum ether, 20 mL was added to the crude extract to remove the chlorophyll from the solvent crude extract and centrifuged at 5000× *g* for 20 min at 4 °C. Then, 100 mL of isopropanol was added to dissolve the crude extract to be tested. Prepared ethanol standard solutions with different concentration gradients of nigranoic acid, 0.2 mL of 5% vanillin glacial acetic acid solution, and 1 mL of perchloric acid were added to 1 mL standard solutions, then placed in a 60 °C water bath for 15 min. After 10 min, 5 mL glacial acetic acid was added, then the absorbance value at 545 nm was measured and the standard curve was drawn. The absorbance value of the prepared isopropanol solution was measured by a similar method, and the total triterpenoid content was calculated according to the standard curve.

For determining flavonoid and polyphenol content, 1 g of plant dried powder was ground with 50 mL 50% ethanol, then transferred into centrifuge tubes and centrifuged at 5000× *g* for 20 min at 4 °C. Then, the crude extract of flavonoids and polyphenols was obtained through the rotation–evaporation method. The standard curve of total flavonoids was drawn with rutin as the standard, and the content of total flavonoids in the crude extract was determined by the classical NaNO_2_–AlCl_3_ spectrophotometry [46]. The standard curve of total polyphenols was drawn with gallic acid as the standard, and the content of total polyphenols in the crude extract was determined by classical Na_2_CO_3_ Folin phenol spectrophotometry [47]. 

### 2.10. Statistical Analysis

All graphing and statistical analyses were performed using Origin Pro 2010 and all the heat maps were drawn with TBtools software. Data were statistically analyzed using the SPSS software version 19.0. Data in the text and tables are expressed as standard deviation means, and error bars in the figures indicate standard deviations. All statistical tests were subjected to ANOVA, the Wilcoxon signed rank test, and least-significant-difference using the *p* < 0.05 significance level.

## 3. Results

### 3.1. Isolation of Endophytic Bacteria and Fungi from S. sphenanthera

A total of 127 bacterial and 119 fungal strains were isolated from the roots, stems, leaves, and fruits of *S. sphenanthera* (see Appendix A). Of these isolates, 27 bacterial and 26 fungal isolates belonging to diverse taxa were chosen for subsequent experiments (see Appendix A). These bacterial isolates represented four phyla, namely, Proteobacteria, Bacteroides, Actinobacteria, and Firmicutes. Fungal isolates represented three phyla, namely, Ascomycota, Basidiomycota, and Zygomycota. These bacterial and fungal strains identified from plant-associated organs are shown in Table 1. 

### 3.2. Isolation and Identification of Triterpenoids from S. sphenanthera 

Four known schitriterpenoids (S1 and S2) and highly oxygenated schinortriterprnoids (S3 and S4), namely, iso-anwuweizicacid (S-1; C_30_H_50_O_2_), nigranoic acid (S2; C_30_H_46_O_4_), micrandilactone C (S3; C_29_H_34_O_9_), and lancifodilactone D (S4; C_29_H_42_O_9_), were obtained from the crude acetone extracts of dry plant organs from *S. sphenanthera* after successive separation and purification using various column chromatography methods (Figure 1). The structures of the known triterpenoids (S1–S4) were identified by comparing the NMR and MS data with those reported in the literature (Appendix A) [48,49,50,51].

### 3.3. Effect of Triterpenoid Metabolites on the Growth of Endophytes 

In order to evaluate the regulative effects of triterpenoid mixtures on the growth of endophytic strains, representative endophytic strains were cultured in a medium containing different initial concentrations of mixed triterpenoid metabolites and their growth was measured. Results indicated that the type of regulation, be it promotion or inhibition, mainly depended on the examined endophytic species and the involved concentrations of triterpenoid mixtures (Figure 2). All examined Actinobacteria isolates were grouped into TNB (bacteria growth-inhibited by triterpenoids), including *Kocuria* sp., *Micrococcus* sp. and *Nocardioides* sp. By contrast, four out of five Bacteroidetes isolates were grouped into TPB, which were *Flavobacterium* sp., *Sphingobacterium* sp., *Alloprevotella* sp., and *Mucilaginibacter* sp. However, for isolates belonging to Firmicutes, five out of eight Firmicutes isolates were grouped into TPB. For isolates belonging to Proteobacteria, only four out of eleven Proteobacteria isolates were grouped into TPB (Figure 2A and Appendix A). Similarly, growth regulatory effects of triterpenoid mixtures were found in fungal strains (Figure 2B and Appendix A). A relatively equal number of isolates belonging to Ascomycota and Basidiomycota were grouped into TPF (fungi growth-promoted by triterpenoids) and TNF (fungi growth-inhibited by triterpenoids). Eight of fifteen Ascomycota isolates were grouped into TPF, and three out of six Basidiomycota isolates were grouped into TPF.

Two trends were observed in the growth-inhibited effect of triterpenoids on TNB strains, one of which was that the growth of TNB strains was progressively enhanced with the increase in mix concentration, such as in *Straphylococcus* sp., *Bacillus* sp. and *Brevibacterium* sp. The other was slightly promoted by low concentrations of the mix (20 and 50 µM) and significantly inhibited by medium to high concentrations of the mix (100 and 250 µM), such as in *Enterobacter* sp., *Nocardioides* sp. and *Straphylococcus* sp. (Figure 2A and Appendix A). By contrast, the growth-inhibited effect of triterpenoids on TNF strains gradually increased with the increase in mix concentration, such as in *Trichoderma* sp., *Penicillium* sp., and *Colletotrichum* sp. (Figure 2B and Appendix A). Overall, triterpenoids showed selective growth regulation to endophytic bacteria and fungi.

### 3.4. Functional Screening of Endophytes 

The representative endophytic strains of diverse taxa were screened with plant growth promotion traits (PGPT). Among bacterial genera, Indole acetic acid (IAA) production tended to be stronger in the bacterial isolates of genera *Sphingobacterium* sp. (66.3 mg/L) and *Kocuria* sp. (66.3 mg/L) than in other isolates. Contrarily, isolates of *Straphylococcus* sp. and *Enterococcus* sp. did not produce IAA. All tested isolates exhibited antioxidative activity, and the isolate with the highest DPPH radical scavenging capacity was the genus of *Enterococcus* sp. (89.7%). Production of P-solubilization was particularly high in genera *Serratia* sp. (137.8 mg/L) and *Lysinibacillus* sp. (129.7 mg/L). *Enterobacter* sp. and *Bacillus* sp. were also strong producers of siderophores, whereas most other tested isolates produced only trace amounts of or no siderophores (see Appendix A). For fungi, the overall share of IAA-producing fungi in our study was also much lower compared to that of bacteria. Isolates of *Lasiodiplodia* sp. (19.1 mg/L) displayed the highest IAA production. We also found that *Penicillium* sp. (175.6 mg/L) and *Perenniporia* sp. (175.6 mg/L) were the most prominent P-solubilizing producers, followed by *Phlebiopsis* sp. (159.4 mg/L) and *Irpex* sp. (157.6 mg/L). Fungal isolates exhibited high antioxidative activity, with 7 of 26 tested fungal isolates at the level of ≥90%. *Penicillium* sp. (53.5 mg/L) and *Pestalotiopsis* sp. (51.0 mg/L) were the strongest producers of siderophores (see Appendix A).

Combining PGPT with the growth-regulating effect of triterpenoids on these endophytes indicated that the PGPT ability of TPB and TPF was significantly higher than that of TNB and TNF. Functional overlaps were common in these microbes, with most TPB and TPF strains possessing one, two, or three PGPT abilities (Figure 3A,B).

### 3.5. Plate Interaction Showed TPB and TPF Are More Important in the Interaction Network

To understand whether and how these TPB and TPF strains might modulate microbial communities in Schisandraceae plants, we also tested 1951 binary interactions in vitro with diverse taxa of 27 bacterial and 26 fungal isolates from our culture collection. The growth of most bacteria was promoted by the bacteria belonging to the isolates of *Bacillus* sp., *Burkholderia* sp., *Sphingobacterium* sp., *Mucilaginibacter* sp., and *Flavobacterium* sp. On the contrary, the growth of most bacteria was inhibited by bacteria belonging to the isolates of *Rhizobium* sp., *Agrobacterium* sp., *Herbaspirillum* sp., *Micrococcus* sp., *Enterobacter* sp., *Limnobacter* sp., and *Duganella* sp. (Figure 4A and Appendix A). Several fungi, such as *Neopestalotiopsis* sp., *Umbelopsis* sp., *Fusarium* sp., and *Phlebia* sp., promoted most of the other fungi, yet fungi belonging to *Trichoderma* sp., *Ceratobasidiu**m* sp., *Lasiodiplodia* sp., and *Clitopilus* sp. inhibited most other fungi (Figure 4B and Appendix A). However, inhibited effects existed between most bacteria and fungi, including several bacterial strains belonging to the *Bacillus* sp., *Viridibacilus* sp., *Pantoea* sp., and *Duganella* sp., which exhibited high inhibitory effects on other fungal isolates (Figure 4C and Appendix A). Above all, the binary interaction phenotypes determined in the microbial interaction screen revealed that most mutual inhibition occurs between bacteria and fungi, and cooperation in the bacterial and fungal communities occurs more than competition.

Combining the interaction strength with the growth-regulating effect of triterpenoids on these endophytes indicated that TPB strains were more promoted by the bacterial community than TNB strains (Figure 5A). However, no significant differences were observed between TPF and TNF strains (Figure 5B). TPF strains *Trichoderma* sp., *Lasiodiplodia* sp., *Clitopilus* sp., and *Ceratobasidium* sp. significantly inhibited the growth of other endophytic fungi (Appendix A). Similar to bacterial results, less inhibition from the bacterial community was observed in TPF strains compared with TNF strains (Figure 5C). These results indicated that triterpenoids could promote the endophytic strains that show more dominance in community interactions.

### 3.6. Evaluation of SynComs

To further investigate the performance of the microbial consortium promoted by triterpenoids, pot experiments were carried out in the growth chambers. By using microbes that were diverse taxa of 27 bacterial and 26 fungal isolates from our culture collection, we constructed six SynComs (synthetic microbial communities) consisting of 27 bacteria (AB), 26 fungi (AF), 27 bacteria and 26 fungi (ABF), 14 growth-inhibited bacteria and 15 growth-inhibited fungi by triterpenoid (TNB&TNF), 13 growth-promoted bacteria and 11 growth-promoted fungi by triterpenoid (TPB&TPF), or control check (CK). In this test, SEM results showed that a large number of endophytic bacteria and endophytic fungi were observed in the roots of TPB&TPF plants compared with CK plants (Figure 6), which proved that the co-culture system was effective for subsequent study. We then explored whether the SynComs could influence plant growth and metabolic accumulation.

Plants inoculated with AB, AF, and ABF significantly promoted *S. sphenanthera* growth, as indicated by a 98.96%, 82.02%, and 115.00% increase in root length, and a 18.45%, 4.49%, and 42.55% increase in plant height, respectively, in comparison with the control plants (Figure 7B). In addition, in comparison with groups AB and AF, the ABF group had significant effects on the growth of *S. sphenanthera*, indicating that endophytic bacteria and fungi had synergistic effects in promoting the growth of *S. sphenanthera*. Moreover, the group of TPB&TPF further promoted the growth of plants, as indicated by a 203.98% and 61.98% root length and plant height increase, respectively, in comparison the with ABF group, whereas the group of TNB&TNF showed insignificant improvement in the growth of plants, as indicated by a 5.73% increase and a 10.3% reduction in root length and plant height, respectively, in comparison with the ABF group (Figure 7B). Further experimentation with SynComs TPB&TPF and TNB&TNF suggested that the microbes promoted by triterpenoids could promote plant growth by increasing root length and plant height. In addition, plants inoculated with AF had a wilt pathogen in the leaves (Figure 7A), as indicated by a 42.18% decrease in the chlorophyll content of the leaves (Figure 8A), indicating that the existence of endophytic bacteria is helpful in reducing the toxic effect of endophytic fungi on plants.

Polyphenols, flavonoids, total sugars, and total triterpenoids are the main metabolites in *S. sphenanthera*. Here, the content of metabolites was determined to evaluate the effects of six SynComs on the metabolic accumulation of *S. sphenanthera*. The polyphenol content increased by 11.12%, 28.37%, 13.76%, 8.56%, and 56.66%; the flavonoid content increased by 9.16%, 58.13%, 25.00%, 8.60%, and 102.21%; the total sugar content increased by 27.66%, 17.87%, 62.91%, 24.75%, and 103.96%; the total triterpenoid content increased by 66.97%, 49.13%, 78.78%, 68.51%, and 240.54% for SynCom AB, AF, ABF, TNB&TNF, and TPB&TPF-treated hosts, respectively (Figure 8B). Results show that all five SynComs could promote metabolite accumulation in the host plants, among which SynCom TPB&TPF accumulated significantly more metabolites of *S. sphenanthera* than the four other SynComs.

## 4. Discussion

Secondary metabolites are a promising target for rationally manipulating the plant microbiome because these molecules frequently exhibit the function of regulating microbes [12]. Specifically selected microbial groups often have special physiological functions for plants. Microbial groups that are specifically regulated by these compounds play an important role in the growth, nutrient absorption, and defense of the host plant [8,9,10,11]. Recent reports have demonstrated that plant secondary metabolites can affect and regulate rhizosphere microbial composition through rhizosphere secretion, including triterpenoids, glucosinolates, flavonoids, phytoalexin, benzoxazines, and coumarins. The mutualistic relationship between plant and endophyte is due to the adaptation and utilization of host secondary metabolites by the endophyte. Thus, we speculated that secondary metabolites were important factors for the enrichment of the microbial community.

Since *S. sphenanthera* is an important medicinal plant that can produce diverse schitriterpenoids [26,27], we tested the hypotheses that triterpenoid-rich medicinal plant *S. sphenanthera* could recruit beneficial endophytic bacteria and fungi through triterpenoids. In our research for the triterpenoid-mediated effects on endophytic microbiome assembly, in vitro traits of the culturable endophytes in *S. sphenanthera* using different concentrations of triterpenoids confirmed these triterpenoids had selective activity on endophytes. First, at the phylum level, the in vitro growth of most Firmicutes, Bacteroides, and some Proteobacteria, Ascomycota, and Basidiomycota was generally promoted by triterpenoids, whereas the growth of most Actinobacteria, and some Proteobacteria, Ascomycota, and Basidiomycota was generally inhibited by triterpenoids. This finding demonstrated that triterpenoids play an important regulatory role in endophytic communities of *S. sphenanthera*. Moreover, the endophytic fungus *Penicillium* sp., which was significantly promoted by triterpenoids, was able to convert and modify host triterpenoids in our previous findings [28]. Previous studies showed that endophytic fungi from *Armoracia rusticana* could decompose glucosinolates by utilizing them as a carbon source [52]. Another study on *Cephalotaxus harringtonia* also showed that endophytic fungus *Paraconiothyrium variabile* could transform the host metabolites of glycosylated flavonoids, and the deglycosylated flavonoids displayed beneficial effects on the hyphal growth of germinated spores [53]. This suggested that the endophytes show adaptation to the host plant’s secondary metabolites and are controlled by plant metabolites. Therefore, we speculated that the regulatory relationship of triterpenoids to endophytes is due to the utilization and metabolism of triterpenoids by endophytes. 

Our results based on the analysis of the interaction between endophytic bacteria and fungi in vitro showed that more mutual promotion existed within the group of bacteria and within the group of fungi, and more mutual inhibition existed between the two groups. These results suggest that the whole endophytic microbial community maintained a relatively balanced state. Similarly, a previous study on the root microbiome of *Arabidopsis thaliana* suggested that filamentous fungi are harmful to the health and survival of plants without bacterial competitors [41]. Bacteria of many genera had significant antagonism to endophytic fungi; species such as *Bacillus* sp., *Viridibacillus* sp., *Pantoea* sp., *Duganella* sp., and *Sphingobacterium* sp. display the strongest antagonistic ability against fungal species. Consistently, these genera could protect plants from pathogens [54,55,56,57]. These results provide a reasonable framework to explain why these strains are widely used in plant biological control in agricultural environments. Some species of *Pantoea* sp. have long been recognized as phytopathogens that can cause diseases of crops and forest trees, whereas some endophytic *Pantoea* sp. show antagonistic abilities against some plant pathogens [58]. This suggests that strains belonging to the same genus may play different roles in various host plants and environments. In total, the balance of endophytic microbial groups in *S. sphenanthera* is not determined by the interaction of a single class, but by the interaction of multi-kingdom microbial consortia, which is conducive to the healthy growth of plants.

Great attention is paid to the endophytes, as they contribute to plant fitness and productivity by providing a plethora of functional capacities, and there is increasing interest in using synthetic communities of microbial strains to promote plant growth fitness and metabolites. Previous studies on soybean plants suggested that constructed synthetic microbial communities based on functional screening were capable of enhancing nutrient acquisition and crop yield through the activities of beneficial root-associated microbes [14]. Currently, these are few guiding principles for constructing SynComs. These include methods to construct synthetic communities according to plant-growth-promotion traits [14], the functional features of individual strains [39], and OTU screening based on high-throughput sequencing [22]. The relationship between secondary metabolites and the microbiome can provide a new tool for the rational manipulation of the plant microbiome [11]. In our study, we considered the effects of triterpenoids on strains and the balance between bacteria and fungi for the construction of SynComs. We grew *S. sphenanthera* plants in soils preinoculated with the six SynComs described in the results to test for their ability to affect plant growth and health. Our results suggest that the inoculation of plants with bacterial communities significantly reverses fungi’s harm for the plant to promote plant growth, health, and metabolic accumulation. Microbial consortia composed of TPB and TPF could significantly promote plant growth, root development, and metabolic accumulation, suggesting that they also synergistically promote the growth of host plants. Among TPB and TPF isolates, some species of *Lysinibacillus* sp., *Brevibacterium* sp., *Colletotrichum* sp., and *Clitopilus* sp. were confirmed to be members of plant-growth-promoting strains [2,58,59,60]. *Ceratobasidium* sp. was capable of facilitating metabolic accumulation in host plants, such as the medicinal plant *Anoectochilus roxburghii* [61]. Some endophytic *Bacillus* sp., *Xanthomonas* sp., *Pantoea* sp., *Trichoderma* sp., and *Fusarium* sp. were treated as an effective bio-control agent against phytopathogenic fungi [20,54,55,56,62,63,64,65,66,67,68]. As a consortium, the microbes promoted by triterpenoids are beneficial to the plant, as they can unite the inter-kingdom and the outer-kingdom microbial consortia, and also promote plant growth. These findings together show that *S. sphenanthera* rich in triterpenoids could recruit specific members of their microbes to aid in their growth, health, and metabolic accumulation. Our results suggest that it is a feasible method for plant growth and metabolism accumulation to construct synthetic communities of bacteria and fungi based on basis of triterpenoid selection.

## 5. Conclusions

We isolated endophytic bacteria and fungi from the plant organs of *S. sphenanthera*, which were assayed by the purified triterpenoids and filtered to construct SynComs on the basis of triterpenoid screening. Our work suggests that triterpenoids, other than their wide biological activity, may play a role in the assembly and stabilization of the plant’s endophytic microbiome. Moreover, the SynComs of endophytes based on the screening of triterpenoids could significantly promote the growth and metabolic accumulation of plants. Our findings demonstrate that triterpenoids play an important regulatory role in endophytic communities of *S. sphenanthera*. Furthermore, these findings provide insights into how plant secondary metabolites affect their endophytic microbial community and how the endophytic bacteria and fungi coexist in plants, providing new approaches to construct SynCom in *S. sphenanthera* which may open new avenues towards the breeding of high-yielding and high metabolite-accumulating medicinal plants by exploiting their interaction with beneficial endophytes. 

## Figures and Tables

**Figure 1 jof-07-00788-f001:**
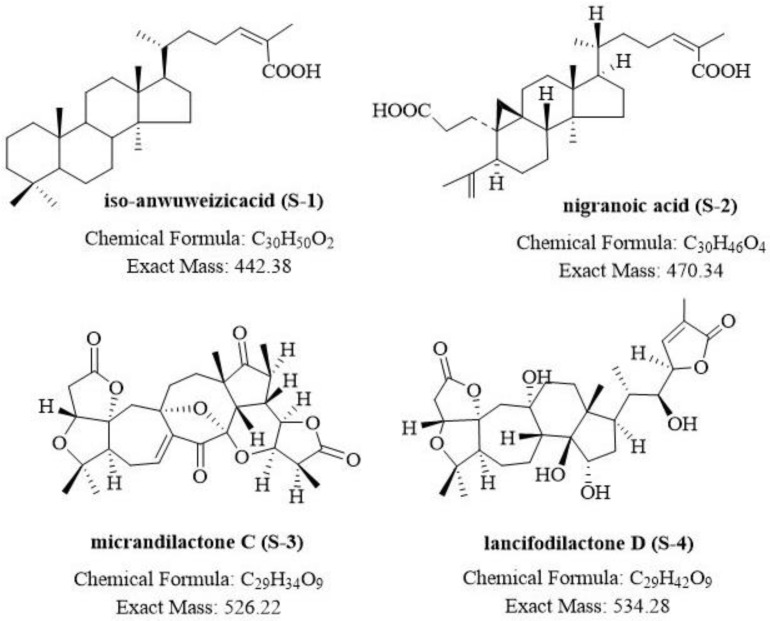
Structures of Schitriterpenoids and Schnortriterpenoids.

**Figure 2 jof-07-00788-f002:**
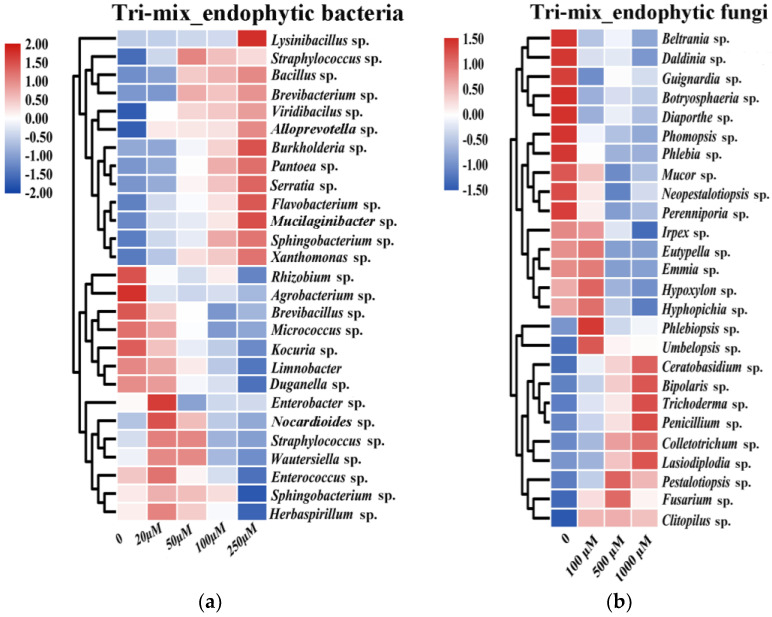
Effects of Triterpenoids on the growth of bacterial and fungal isolates. Heatmap showing log2 fold change in (**a**) cell density (OD_600_) of bacterial strain and (**b**) weight (g) of fungal strain treated with mix (S1–4) by gradient concentration at 48 h (bacteria) and 12 days (fungi). Log2 fold change is an estimate of the log2 ratio of relative value to that initial value. A value of 1.0 indicates two-fold greater expression in heatmap cluster. Five biological replicates per treatment were used for analysis.

**Figure 3 jof-07-00788-f003:**
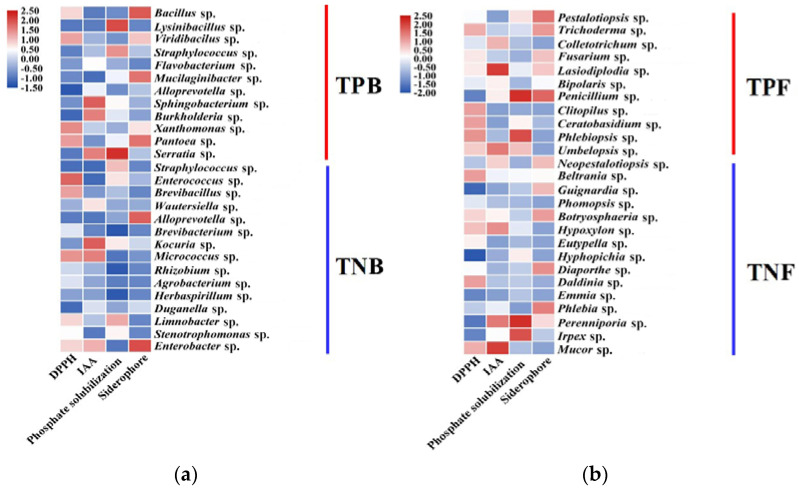
Plant growth promotion traits (PGPT) and triterpenoid synthesis capacity of representative bacterial and fungal isolates. (**a**) Heatmap of PGPT of 27 endophytic bacterial isolates. (**b**) Heatmap of PGPT of 26 endophytic fungal isolates. Three biological replicates per treatment were used for analysis.

**Figure 4 jof-07-00788-f004:**
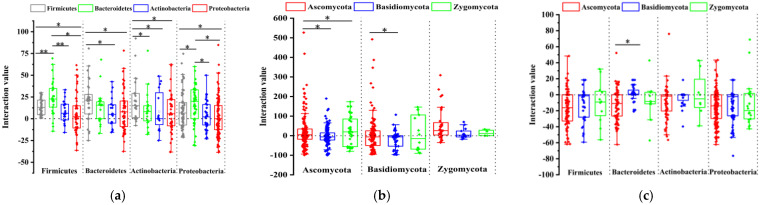
Interactions of representative bacterial and fungal isolates. (**a**) Box-plot analysis of endophytic bacterial interactions at the taxonomic level of the phylum. (**b**) Box-plot analysis of endophytic fungal interactions at the taxonomic level of the phylum. (**c**) Box-plot analysis of the interaction between endophytic bacteria and fungi at the taxonomic level of the phylum. Asterisks indicate significant differences between the group: ** *p* < 0.01, * *p* < 0.05, nonparametric test, Wilcoxon signed rank test. Three biological replicates per treatment were used for analysis. Error bars represent the SEM.

**Figure 5 jof-07-00788-f005:**
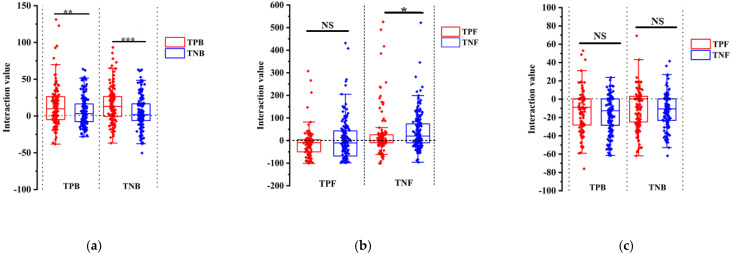
Correlation analysis between growth regulation effect of triterpenoids and the interaction. (**a**) Correlation analysis between growth regulation effect of triterpenoids on endophytic bacteria and the interaction of endophytic bacteria. (**b**) Correlation analysis between growth regulation effect of triterpenoids on endophytic fungi and the interaction of endophytic fungi. (**c**) Correlation analysis between growth regulation effect of triterpenoids on endophytes and the interaction between endophytes. Asterisks indicate significant differences between the group: *** *p* < 0.001, ** *p* < 0.01, * *p* < 0.05, nonparametric test, Wilcoxon signed rank test. Three biological replicates per treatment were used for analysis. Error bars represent the SEM.

**Figure 6 jof-07-00788-f006:**
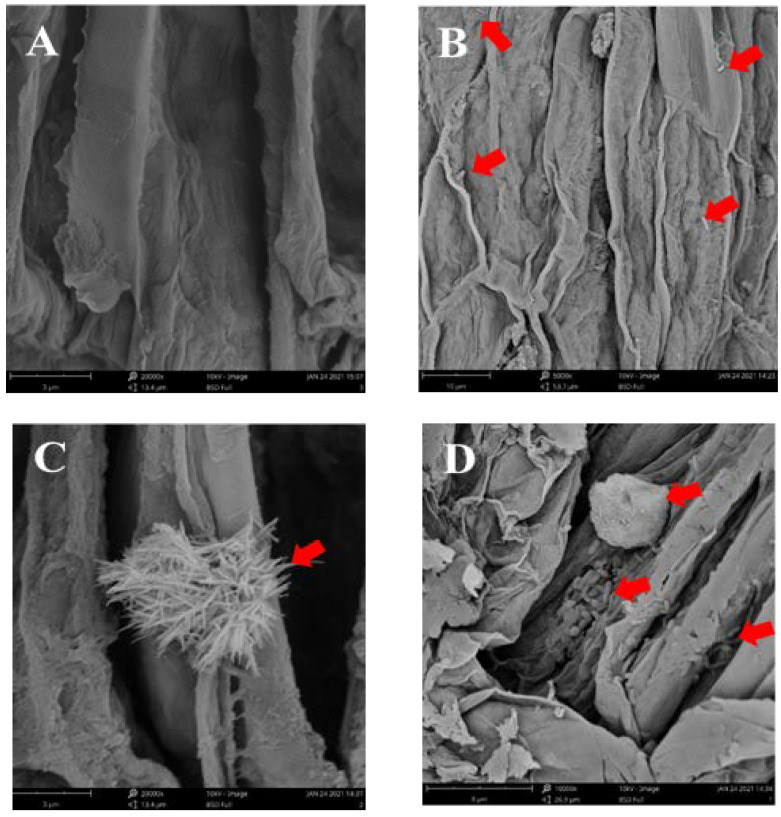
Colonization of *S. sphenanthera* roots by bacteria and fungi in different treatment groups. (**A**) Roots of *S. sphenanthera* in control group, in which no microbes were found. (**B**) Roots of *S. sphenanthera* in AB, in which red arrow represents colonized bacteria. (**C**) Roots of *S. sphenanthera* in AF, in which red arrow represents colonized fungi. (**D**) Roots of *S. sphenanthera* in ABF group, in which red arrow represents coexistence of colonized bacteria and fungi.

**Figure 7 jof-07-00788-f007:**
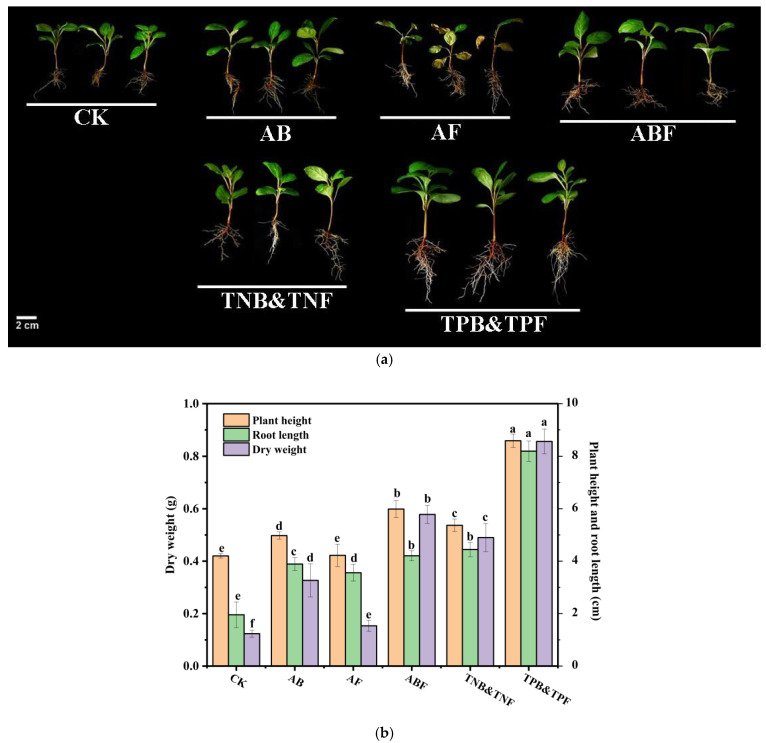
Pot experiment of SynCom in germ-free *S. sphenanthera* plants. (**a**) Growth status of *S. sphenanthera* in different treatment groups. (**b**) Effects of different SynComs on root length, stem length, and dry weight of *S. sphenanthera*. Different lowercase letters (a–f) indicate significant differences between different treatments, one-way ANOVA, Duncan’s test. Error bars represent the SEM.

**Figure 8 jof-07-00788-f008:**
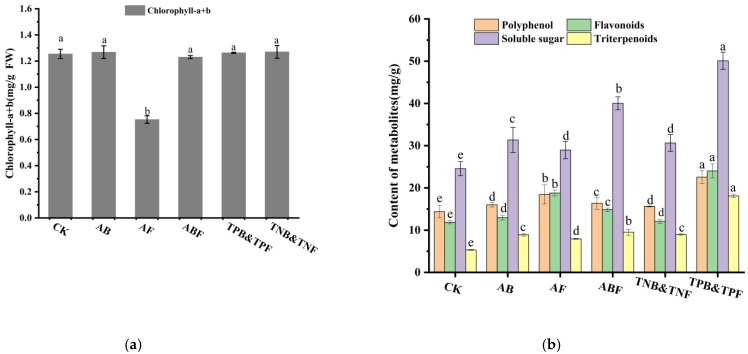
Effects of different SynComs on the contents of metabolites in *S. sphenanthera*. (**a**) Effects of different SynComs on contents of total sugar, flavonoids, polyphenols, and total terpenoid in *S. sphenanthera*. (**b**) Effects of different SynComs on the chlorophyll content of *S. sphenanthera*. Different lowercase letters (a–e) indicate significant differences between different treatments, one-way ANOVA, Duncan’s test. Error bars represent the SEM.

**Table 1 jof-07-00788-t001:** Representative endophytic bacterial and fungal strains isolated from *S. sphenanthera*.

Bacteria			Fungi		
Phylum	Genus	Numbers	Phylum	Genus	Numbers
Firmicutes	*Bacillus* sp.	SLB12	Ascomycota	*Pestalotiopsis* sp.	SRF21
Firmicutes	*Lysinibacillus* sp.	SRB6	Ascomycota	*Trichoderma* sp.	SSF13
Firmicutes	*Viridibacilus* sp.	SFB11	Ascomycota	*Colletotrichum* sp.	SSF14
Firmicutes	*Romboutsia* sp.	SRB23	Ascomycota	*Fusarium* sp.	SFF11
Firmicutes	*Straphylococcus* sp.	SSB39	Ascomycota	*Lasiodiplodia* sp.	SLF1
Firmicutes	*Enterococcus* sp.	SLB1	Ascomycota	*Bipolaris* sp.	SSF12
Firmicutes	*Brevibacillus* sp.	SFB12	Ascomycota	*Penicillium* sp.	SRF9
Bacteroidetes	*Flavobacterium* sp.	SSB1	Ascomycota	*Diaporthe* sp.	SLF18
Bacteroidetes	*Sphingobacterium* sp.	SRB12	Ascomycota	*Daldinia* sp.	SLF4
Bacteroidetes	*Mucilaginibacter* sp.	SSB4	Ascomycota	*Neopestalotiopsis* sp.	SLF5
Bacteroidetes	*Wautersiella* sp.	SSB19	Ascomycota	*Beltrania* sp.	SLF21
Bacteroidetes	*Alloprevotella* sp.	SLB9	Ascomycota	*Guignardia* sp.	SLF6
Actinobacteria	*Nocardioides* sp.	SSB16	Ascomycota	*Phomopsis* sp.	SFF9
Actinobacteria	*Micrococcus* sp.	SLB31	Ascomycota	*Botryosphaeria* sp.	SRF16
Actinobacteria	*Brevibacterium* sp.	SLB15	Ascomycota	*Hypoxylon* sp.	SLF24
Actinobacteria	*Kocuria* sp.	SLB14	Ascomycota	*Eutypella* sp.	SSF11
Proteobacteria	*Pantoea* sp.	SSB8	Ascomycota	*Hyphopichia* sp.	SFF7
Proteobacteria	*Burkholderia* sp.	SRB18	Basidiomycota	*Clitopilus* sp.	SSF19
Proteobacteria	*Serratia* sp.	SFB13	Basidiomycota	*Ceratobasidium* sp.	SFF6
Proteobacteria	*Xanthomonas* sp.	SFB7	Basidiomycota	*Phlebiopsis* sp.	SSF18
Proteobacteria	*Limnobacter* sp.	SLB7	Basidiomycota	*Emmia* sp.	SFF12
Proteobacteria	*Duganella* sp.	SRB22	Basidiomycota	*Irpex* sp.	SRF17
Proteobacteria	*Rhizobium* sp.	SLB15	Basidiomycota	*Phlebia* sp.	SLF27
Proteobacteria	*Agrobacterium* sp.	SSB31	Basidiomycota	*Perenniporia* sp.	SLF26
Proteobacteria	*Herbaspirillum* sp.	SFB6	Zygomycota	*Umbelopsis* sp.	SLF28
Proteobacteria	*Stenotrophomonas* sp.	SSB25	Zygomycota	*Mucor* sp.	SRF1
Proteobacteria	*Enterobacter* sp.	SLB11			

## Data Availability

Genome accession numbers for the bacterial and fungal strains sequenced in this study were deposited in GenBank under accession numbers MZ707882-MZ707908 and MZ710145-MZ710170.

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
