# Peer review of "Plant Triterpenoids Regulate Endophyte Community to Promote Medicinal Plant Schisandra sphenanthera Growth and Metabolites Accumulation"

_jof, 2021, doi:10.3390/jof7100788_

Round 1

Reviewer 1 Report

Dear Authors,

Please correct the following minor points.

Minor points (scientific)

L118: 16S rRNA or ITS rDNA (not "and")

L130: add additional params to MCI (should be like: "MCI GEL CHP20P")

L363&L371: The claims regarding triterpene content of microbes should be removed, I think these are accidently left here. Correctly, there is no mentioning of application of triterpenoid assay in microbes in the M&M section. As I noted in a previous review, the procedure used for plants is not suitable to measure triterpenoids in microbes due to low selectivity of the used method.

Minor points (technical and English)

L17: metabolite accumulation

L28: role of triterpenoids

L31: plants by

L172: define chrome azurol S here and not later

L201: the equation subscripts should be "mm" not "cm"

L225: MS + BA (1.0 mg/L) + NAA (0.03 mg/L)

L273: Of pure pethroleum ether, 20 ml was added ...

L345: Overall, triterpenoids ...

Fig3: letters at the right side of the subplots are cut off, please fix this figure

L394: high inhibitory effects on

L501: could transform

Best regards.

Author Response

Thanks a lot for reviewing our paper “Plant triterpenoids regulate endophyte community to promote medicinal plant Schisandra sphenanthera growth and  metabolites accumulation”. We would appreciate you for giving us an opportunity to revise and resubmit our manuscript. Your comments are all valuable and very helpful for revising and improving our paper, as well as the important guiding significance to our research. We have studied the comments carefully and have made minor revision which emphasis on the text in red font. We have tried our best to revise our manuscript according to the comments.

Minor points (scientific)

1.Comment: 

L118: 16S rRNA or ITS rDNA (not "and")

1.Reply: this mistake was corrected in line 116

2.Comment: L130: add additional params to MCI (should be like: "MCI GEL CHP20P")

2.Reply: We have added additional params of MIC in line 132.

3.Comment: L363&L371: The claims regarding triterpene content of microbes should be removed, I think these are accidently left here. Correctly, there is no mentioning of application of triterpenoid assay in microbes in the M&M section. As I noted in a previous review, the procedure used for plants is not suitable to measure triterpenoids in microbes due to low selectivity of the used method.

3.Reply: Thanks for your reminder. We have removed the wrong section.

Minor points (technical and English)

1.Comment: 

L17: metabolite accumulation

1.Reply: “metabolic accumulation” have been corrected into “metabolite accumulation” in line 17.

2.Comment: L28: role of triterpenoids

2.Reply: “role for triterpenoid” have been corrected into “role of triterpenoids” in line 28.

3.Comment: L31: plants by

3.Reply: “plant by” have been corrected into “plants by” in line 31.

4.Comment: L172: define chrome azurol S here and not later

4.Reply: we have defined chrome azurol S in line 174.

5.Comment: L201: the equation subscripts should be "mm" not "cm"

5.Reply: Thanks for your reminder. “cm” have been corrected into “mm” in line 203.

6.Comment: L225: MS + BA (1.0 mg/L) + NAA (0.03 mg/L)

6.Reply: This mistake has been corrected in line 228.

7.Comment: L273: Of pure pethroleum ether, 20 ml was added ...

7.Reply: The redundant part has been deleted in line 278.

8.Comment: L345: Overall, triterpenoids ...

8.Reply: “In total” has been corrected into “Overall” in line 352.

9.Comment: Fig3: letters at the right side of the subplots are cut off, please fix this figure

9.Reply: Thanks for your reminder. We have fixed figure 3.

10.Comment: L394: high inhibitory effects on

10.Reply: “high inhibited effects” has been corrected into “high inhibitory effects” in line 401.

11.Comment: L501: could transform

“could transformed” has been corrected into “could transform” in line 511.

Reviewer 2 Report

The manuscript has been revised and improved as per my suggested comments in the last round of review 

Author Response

Thanks a lot for reviewing our paper “Plant triterpenoids regulate endophyte community to promote medicinal plant Schisandra sphenanthera growth and  metabolites accumulation”. We would appreciate you for giving us an opportunity to revise and resubmit our manuscript. Your comments are all valuable and very helpful for revising and improving our paper, as well as the important guiding significance to our research. We have studied the comments carefully and have made minor revision which emphasis on the text in red font. We have tried our best to revise our manuscript according to the comments.

This manuscript is a resubmission of an earlier submission. The following is a list of the peer review reports and author responses from that submission.

Round 1

Reviewer 1 Report

Review Comments - This manuscript demonstrates that plant-derived triterpenoids predominantly promote the growth of endophytic bacteria and fungi, which in turn promote host plant growth and secondary metabolite productions. The findings may open new avenues towards the breeding of high yielding and high metabolite-accumulating medicinal plant by exploiting their interaction with beneficial endophytes. The study is well conducted and provide useful information to the readers. However, some revisions are still required as shown below; - Methods should be written in more detail. - The discussion should be interpreted with the results as well as discussed in relation to the present literature. - A conclusion section should highlight the important findings of this study. - English language of the whole manuscript has to be improved and corrected by native English speaker or English proofreading service.

Author Response

Thanks a lot for reviewing our paper “Plant triterpenoids regulate endophyte community to promote medicinal plant Schisandra sphenanthera growth and  metabolites accumulation”. We would appreciate you for giving us an opportunity to revise and resubmit our manuscript. Your comments are all valuable and very helpful for revising and improving our paper, as well as the important guiding significance to our research. We have studied the comments carefully and have made major revision which emphasis on the text in red font. We have tried our best to revise our manuscript according to the comments.

Comment: This manuscript demonstrates that plant-derived triterpenoids predominantly promote the growth of endophytic bacteria and fungi, which in turn promote host plant growth and secondary metabolite productions. The findings may open new avenues towards the breeding of high yielding and high metabolite-accumulating medicinal plant by exploiting their interaction with beneficial endophytes. The study is well conducted and provide useful information to the readers. However, some revisions are still required as shown below; - Methods should be written in more detail. - The discussion should be interpreted with the results as well as discussed in relation to the present literature. - A conclusion section should highlight the important findings of this study. - English language of the whole manuscript has to be improved and corrected by native English speaker or English proofreading service.

Reply: We appreciate for your valuable comment. Firtsly, we rewrote the contents of materials and methods and the details of some experiments. At the same time, combined with the latest relevant literature, we have a further in-depth discussion on the content of the discussion part of this paper. The highlight and the important findings of this study also been added to the conclusion section. Additionally, the English language of the whole manuscript has been improved and corrected by the native English speaker. Please confirm the above modifications in the newly revised manuscript.

Reviewer 2 Report

Dear Authors,

There are a few flaws in methodology and data presentation that needs to be addressed before publication. Otherwise, you present a really nice paper with interesting results and an original approach. Please carry out a revision according to the following suggestions:

Major issues (scientific)

1., The spectrophotometric assay for "triterpenoids" (L166-168) has such a low specificity that you cannot get meaningful data out from it when applied to such a variable list of microbes. Possible interferences include steroids and a wide range of other terpenoids, just to mention a few. The data produced with it therefore must be removed (L338-341, Fig.3c and elsewhere), as it does not provide evidence for triterpenoid accumulation in microbes. If Schisandra has this as an accepted quantification method, add a reference or a pharmacopoeia link. Otherwise, remove results from the plant as well. For such complex and variable compounds, LC-UV(-MS) has become the minimum. Your other assays are still accepted as routine measurements as far as I know. This change won't significantly reduce the novelty of your paper.

2., How do you ensure residual water does not interfere with fresh weight measurement of fungi? It can add a terrible amount of noise, as after centrifugation ends, swelling begins immediately. What RSDs did you have within replicates? (Or: what was the typical amount you measured and what kind of weight measurement precision was available?) Use dry weight in the future, whenever possible.

3., Why were a mixture of stem-, leaf- and root derived endophytes mixed together and all injected to soil? Please provide references and details to support this decision.

4., You speak of "selective activity" in L481. I think your results suggest some sort of specific adaptation of endophytes towards the chemical compounds in the plant. Similar conclusion were drawn for other compounds classes, e.g. by 10.1186/s12870-018-1295-4 for glucosinolates and isothiocyanates derived thereof, and the work of 10.1016/j.phytochem.2014.09.021 for flavonoids also points towards this. Add these to your discussion.

Major issues (technical)

The section L382-399 is a very hard read, and it needs revision.

The section L485-493 is a very hard read, and it needs revision. Split into simpler sentences.

Minor issues (scientific)

- Was the 10kg plant source from the same site?

- Who identified the plants?

- Add the amount of acetone used for soaking.

- Add the details of the elution gradient for preparative chromatography. It is unclear what an MCI column is, and which eluting solvent was used for obtaining the needed fractions, and which ones were used to remove pigments and unnecessary compounds.

- What is an aceton gel column? Add the column material.

- L157: This is a Fe2+ assay, insn't it? Why "P"? Perhaps this is a typo.

- L211, L261, L263 and elsewhere: Chemical compound atom numbers should be in subscript.

- L246: triterpenoids, not sugars, right?

- L290: What fermentation?

- What purity were for your isolated standards (LC-UV % at 200 nm for example)?

- Does Fig. 2. show log2 fold change vs initial conditions (0 time)? Consider revising the caption to make it more clear.

- L374: Ceratobasidium

- Fig.4.: The values cannot follow a normal distribution, as the interaction value cannot go below -100, if I understood correctly. Therefore, Kruskal-Wallis test should be used instead of ANOVA. Also remove the bottom part of Fig4b (that below -100), as it is not meaningful.

- Add number of replicates to Figure captions, where statistical tests are mentioned.

Minor issues (technical)

- L39: plants, including

- L43: in the vicinity

- L270: Student's t-test

- L279: perhaps you meant "belonging to diverse taxa"

- L301: bacteria growth-promoted by triterpenoids

- L305: fungi growth-promoted by triterpenoids

- L498: ... existed within the group of bacteria and within the group of fungi, and more mutual inhibition existed between the two groups.

- L537: S. sphenanthera rich in triterpenoids could recruit

- Add manufacturers to chemicals and solvents (supplementary is OK).

- Add a list of abbreviations. You have many unresolved ones.

- Taxonomic names should be in italic throughout the paper.

- I suggest "AF", "AB" and "ABF" (all fungi, all bacteria, all bacteria and fungi) instead of TPF&TNF, TPB&TNB, TPF&TNF&TPB&TNB, respectively.

Author Response

Dear Reviewer,

Thanks a lot for reviewing our paper “Plant triterpenoids regulate endophyte community to promote medicinal plant Schisandra sphenanthera growth and  metabolites accumulation”. We would appreciate you for giving us an opportunity to revise and resubmit our manuscript. Your comments are all valuable and very helpful for revising and improving our paper, as well as the important guiding significance to our research. We have studied the comments carefully and have made major revision which emphasis on the text in red font. We have tried our best to revise our manuscript according to the comments.

Now, we have modified our manuscript point by point according to the comments of reviewers. They mainly include:

Major issues (scientific)

1.Comment: The spectrophotometric assay for "triterpenoids" (L166-168) has such a low specificity that you cannot get meaningful data out from it when applied to such a variable list of microbes. Possible interferences include steroids and a wide range of other terpenoids, just to mention a few. The data produced with it therefore must be removed (L338-341, Fig.3c and elsewhere), as it does not provide evidence for triterpenoid accumulation in microbes. If Schisandra has this as an accepted quantification method, add a reference or a pharmacopoeia link. Otherwise, remove results from the plant as well. For such complex and variable compounds, LC-UV(-MS) has become the minimum. Your other assays are still accepted as routine measurements as far as I know. This change won't significantly reduce the novelty of your paper.

1.Reply:1 We appreciate for your valuable comment. The determination of the total content of triterpenoid was performed as described bySun, Kuo and Yi et al. The total triterpenoids contents of different solvent extracts from Betula, Chinese olive (Canarium album L.) fruit and Ganoderma atrum were determined by spectrophotometric methods.

(Sun, H. et al. 2005. DOI: 10.3969/j.jssn.1000-2006.2005.01.027;

Kuo, CT et al. 2015. DOI: 10.1016/j.apjtm.2015.11. 013;

Yi, Chen et al. 2006. DOI: 10.1016/j. jfoodeng.2006.10.018).

The total content of triterpenoids was determined as described previously with slight modifications. Phytochemical studies have shown that these plants including Betula, Canarium album L. and Schisandra sphenanthera is rich in triterpenoids. We think the spectrophotometric assay for total triterpenoids to the extraction of other phytochemicals from tissue of other medicinal plants is also expected. But these method is not variable for microbes. Thus, we remove results from the microbes in Fig.3c.

2.Comment: 2., How do you ensure residual water does not interfere with fresh weight measurement of fungi? It can add a terrible amount of noise, as after centrifugation ends, swelling begins immediately. What RSDs did you have within replicates? (Or: what was the typical amount you measured and what kind of weight measurement precision was available?) Use dry weight in the future, whenever possible.

2.Reply: Five replicates were used for weight measurement. In order to test the stability of fresh weight, we repeated the determination 12 hours after the first determination, and the results showed that there was no significant difference in weight. Thank you again for your reminder. In future research, we will consider using dry weight instead of fresh weight to evaluate the corresponding experimental results.

3.Comment: 3., Why were a mixture of stem-, leaf- and root derived endophytes mixed together and all injected to soil? Please provide references and details to support this decision.

3.Reply: According the research of Bai et al. (2015, nature. doi: 10.1038/ nature16192), thay established Arabidopsis leaf- and root-derived microbiota culture collections representing the majority of bacterial species. The synthetic bacterial communities (SynComs) were inoculated into the calcined clay matrix before sowing of surface-sterilized seeds. Then, these isolates can still colonize into the corresponding organs in the original host plant. At the same time, our research group has been engaged in the mutual research of plants and endophytes for a long time. This research method has been used in many of our studies and has been recognized by many reviewers and editors. Therefore, we adopted a similar research method in this study.

4.Comment: 4., You speak of "selective activity" in L481. I think your results suggest some sort of specific adaptation of endophytes towards the chemical compounds in the plant. Similar conclusion were drawn for other compounds classes, e.g. by 10.1186/s12870-018-1295-4 for glucosinolates and isothiocyanates derived thereof, and the work of 10.1016/j.phytochem.2014.09.021 for flavonoids also points towards this. Add these to your discussion.

4.Reply: We appreciate for your valuable literstures. Combined with these relevant literatures, further in-depth discussion has been added in lines 514-523.

Major issues (technical)

Comment: The section L382-399 is a very hard read, and it needs revision.

The section L485-493 is a very hard read, and it needs revision. Split into simpler sentences.

Reply: We have rewritten these sections of lines of 409-417 and 564-570 in the newly submitted manuscript.

Minor issues (scientific)

1.Comment: Was the 10kg plant source from the same site?

1.Reply:The roots, stems, leaves and fruits (10 kg) samples from five S. sphenanthera plants in the same location, which were cut into pieces and dried naturally. (line 127)

2.Comment: Who identified the plants?

2.Reply: The plant was identified by Professor Hongping Deng, doctoral supervisor; dean-president of the School of Life Science, Southwest University; Chairman of the Chongqing Plant Society, Director of the China Botanical Society, and Vice President of the Chongqing Wildlife Conservation Society. He is mainly engaged in scientific research work in plant systems, conservation biology, and resource botany.

3.Comment: Add the amount of acetone used for soaking. 

3.Reply: After dried, the plant organs were soaked in 20 L of acetone for 15 days. (line 128)

4.Comment: Add the details of the elution gradient for preparative chromatography. It is unclear what an MCI column is, and which eluting solvent was used for obtaining the needed fractions, and which ones were used to remove pigments and unnecessary compounds.

4.Reply: Reminded by the reviewer, we have re-described in detail the experimental process of separation and purification of the four main triterpenoids in the study and all the information involved in the newly uploaded manuscript. Please check in lines 131 to 144 of the paper.

5.Comment: What is an aceton gel column? Add the column material.

5.Reply: The column material is Sephadex LH-20, we have added the corresponding content in the article.

6.Comment: L157: This is a Fe2+ assay, insn't it? Why "P"? Perhaps this is a typo.

6.Reply: This is a typing error. To evaluate siderophore production, CAS medium was used to determine the siderophore production of the selected strains as described by Alexander et al. (line 174)

7.Comment: L211, L261, L263 and elsewhere: Chemical compound atom numbers should be in subscript.

7.Reply: In your reminder, we added the chemical molecular formulas of four triterpenoids in the article, including iso-anwuweizicacid (S-1; C30H50O2), nigranoic acid (S-2; C30H46O4), micrandilactone C (S-3; C29H34O9) and lancifodilactone D (S-4; C29H42O9) (line 315-317, Fig 1)

8.Comment: L246: triterpenoids, not sugars, right?

8.Reply: this is a typing error, sugar was corrected into triterpenoids in line 276

9.Comment: L290: What fermentation?

9.Reply: This is a typing error, it has been corrected into the crude acetone extracts of the dry plant organs from S. sphenanthera (line 318).

10.Comment: What purity were for your isolated standards (LC-UV % at 200 nm for example)?

10.Reply: Under the recommendation of the reviewer, we added the details of the purity analysis of the three triterpenoids separated from the study in the revised manuscript, including “The purity of all the isolated triterpenoids was determined by HPLC analytsis, which was carried out using an Agilent L-2000 liquid chromatograph equipped with a Zorbax SB-C18 (Agilent Technologies, Sants Clara-California, 9.4 mm × 15 cm column), The detection wavelengths of compounds S1-S4 are 212 nm (LC-UV%), 217 nm (LC-UV%), 203 nm (LC-UV%) and 243 nm (LC-UV%)”. (line 131-144)

11.Comment: Does Fig. 2. show log2 fold change vs initial conditions (0 time)? Consider revising the caption to make it more clear.

11.Reply: The Log2 fold-change is an estimate of the log2 ratio of relative value  to that initial value. A value of 1.0 indicates 2-fold greater expression in the cluster of heatmap. (line 358-359)

12.Comment: L374: Ceratobasidium

12.Reply: This is a type error, Ceratobasidiu was corrected into Ceratobasidium in line 401

13.Comment: Fig.4.: The values cannot follow a normal distribution, as the interaction value cannot go below -100, if I understood correctly. Therefore, Kruskal-Wallis test should be used instead of ANOVA. Also remove the bottom part of Fig4b (that below -100), as it is not meaningful.- Add number of replicates to Figure captions, where statistical tests are mentioned.

13.Reply: Under your reminder, we checked our corresponding analysis interaction value again and found that there was no value lower than -100. At the same time, we have tried the statistical method of Kruskal-Wallis test by the reviewer to re-analyze our results. Unfortunately, this test did not find significant differences between most groups. Therefore, we still adhere to the original statistical analysis method. Compared with Kruskal-Wallis test, the difference between groups of ANOVA test is more significant. Three biological replicates per treatment were used for analysis. (Fig.4)

Minor issues (technical)

1.Comment: L39: plants, including

1.Reply: Endophytic microbiome plays an important role in different functions such as plant growth, development, signal network, gene expression, metabolic pathways and response to biological stress. (line 41)

2.Comment: L43: in the vicinity

2.Reply: live in the vicinity, surface, and inside their tissues. (line 45)

3.Comment: L270: Student's t-test

3.Reply: This is a mistake, Student's t-test were delated in Statistical analysis.

4.Comment: L279: perhaps you meant "belonging to diverse taxa"

4.Reply: 27 bacterial and 26 fungal isolates belonging to diverse taxa were picked for subsequent experiments. (line 307)

5.Comment: L301: bacteria growth-promoted by triterpenoids; L305: fungi growth-promoted by triterpenoids

5.Reply: We have corrected all the mistakes in lines 332-342

6.Comment: L498: ... existed within the group of bacteria and within the group of fungi, and more mutual inhibition existed between the two groups.

6.Reply: ... showed that more mutual promotion existed within the group of bacteria and within the group of fungi, and more mutual inhibition existed between the two groups. (line 525-527)

7.Comment: L537: S. sphenanthera rich in triterpenoids could recruit

7.Reply: Together these findings show that S. sphenanthera rich in triterpenoids could recruit specific members... (line 573)

8.Comment: - Add manufacturers to chemicals and solvents (supplementary is OK).- Add a list of abbreviations. You have many unresolved ones.

8.Reply: Table S4 and S5 is supplemented to illustrate the chemicals, solvents and abbreviations. (Supplementary Materials)

9.Comment: - Taxonomic names should be in italic throughout the paper.

9.Reply: Taxonomic names have been corrected into italic throughout the paper.

10.Comment: I suggest "AF", "AB" and "ABF" (all fungi, all bacteria, all bacteria and fungi) instead of TPF&TNF, TPB&TNB, TPF&TNF&TPB&TNB, respectively.

10.Reply: We appreciate for your valuable suggestion and we have followed your advice to correct it. (lines 436-437, Fig7,8)

Round 2

Reviewer 2 Report

Dear Authors,

Thank you for addressing the points I raised, I only have a few comments left to be dealt with. Please carry out a revision according to the following suggestions.

1. Regarding my previous points/Minor/scientific/13.: Kruskal-Wallis has less statistical power compared to an ANOVA, but to use ANOVA, your data must test so-called ANOVA assumptions (please just google it, there are lots of guides out there). You should not choose it only because it is more powerful (detects differences more easily). Please run these assumptions or you'll have to stick to KW.

2. Purity still not presented in the paper for purified standards. It should be >= 95%.

Minor issues (technical)

Fig.3. subplots: the right edges were cropped, hiding B/F in category names

L340-341: µM not uM

Best regards.

Author Response

Thanks a lot for reviewing our paper “Plant triterpenoids regulate endophyte community to promote medicinal plant Schisandra sphenanthera growth and  metabolites accumulation”. We would appreciate you for giving us an opportunity to revise and resubmit our manuscript. Your comments are all valuable and very helpful for revising and improving our paper, as well as the important guiding significance to our research. We have tried our best to revise our manuscript according to the comments. Now, we have modified our manuscript point by point according to the comments of reviewers. They mainly include:

1. Comment: 1. Regarding my previous points/Minor/scientific/13.: Kruskal-Wallis has less statistical power compared to an ANOVA, but to use ANOVA, your data must test so-called ANOVA assumptions (please just google it, there are lots of guides out there). You should not choose it only because it is more powerful (detects differences more easily). Please run these assumptions or you'll have to stick to KW.

1. Reply:We conducted the hypothesis test (Two Sample Test variance) between two samples in the same group in Figure 4 and found that there were significant differences between all the two samples, whether F-test or Levene's test. Therefore, through this result, we think that the statistical analysis result of one-way ANOVA used in Figure 4 is not applicable. Then, we re-analyzed all the data in Fig. 4 and Fig. 5 by using the paired sample Wilcoxon signed rank test of nonparametric test. Thank you again for your patient guidance on this issue.

2. Comment: 2. Purity still not presented in the paper for purified standards. It should be >= 95%.

2. Reply:Thanks for your reminder.“ The purity of all the isolated triterpenoids was greater than 95%” were added in line 139.

Minor issues (technical)

1. Comment: Fig.3. subplots: the right edges were cropped, hiding B/F in category names

1. Reply:We have cropped the right edges to hide B/F. (Fig.3)

2. Comment: L340-341: µM not uM

2. Reply:We have corrected the mistake in lines L340-341.
